# Pragmatic Reasoning Unlocks Quantifier Semantics for Foundation Models

**Yiyuan Li**      **Rakesh R. Menon**      **Sayan Ghosh**      **Shashank Srivastava**

UNC Chapel Hill

{yiyuanli, rrmenon, sayghosh, ssrivastava}@cs.unc.edu

## Abstract

Generalized quantifiers (e.g., *few, most*) are used to indicate the proportions predicates are satisfied (for example, *some* apples are red). One way to interpret quantifier semantics is to explicitly bind these satisfactions with percentage scopes (e.g., *30%-40%* of apples are red). This approach can be helpful for tasks like logic formalization and surface-form quantitative reasoning (Gordon and Schubert, 2010; Roy et al., 2015). However, it remains unclear if recent foundation models possess this ability, as they lack direct training signals. To explore this, we introduce QuRe, a crowd-sourced dataset of human-annotated generalized quantifiers in Wikipedia sentences featuring percentage-equipped predicates. We explore quantifier comprehension in language models using PRESQUE, a framework that combines natural language inference and the Rational Speech Acts framework. Experimental results on the HVD dataset and QuRe illustrate that PRESQUE, employing pragmatic reasoning, performs 20% better than a literal reasoning baseline when predicting quantifier percentage scopes, with no additional training required[1].

## 1 Introduction

Generalized quantifiers (Mostowski, 1957) are used to express relations between subsets of concepts or entities. For instance, the quantifier '*some*' in the statement '*some apples are red*' indicates that at least one apple is red. Quantifiers, being inherently fuzzy, are prevalent in both real-world communication and natural language processing (NLP) benchmarks (Joshi et al., 2020). Consequently, developing a formal framework for understanding quantifier semantics is essential to enhance the language understanding capabilities of NLP systems, particularly in facilitating natural human-AI language-based interactions, such as in human-robot collaborative tasks (Alami, 2013).

In this work we present PRESQUE (**P**ragmatic **RE**asoning for **S**emantics of **QU**antifi**E**rs), a framework to model the semantics of quantifiers for text-based foundation models, such as BERT (Devlin et al., 2019) and RoBERTa (Liu et al., 2019), , through the lens of pragmatic reasoning. While foundation models have shown impressive performance on various text-based tasks (Bommasani et al., 2021; Wei et al., 2022), their ability to infer the semantic meanings of generalized quantifiers remains relatively unexplored.

In PRESQUE, we represent quantifier semantics in terms of percentage scopes, which indicate the proportion of cases where the associated predicate holds true. For example, in the sentence 'some apples are red', the quantifier 'some' could be associated with a percentage scope of 30-40%, indicating that 30-40% of all apples are red. Our framework consists of two components: (1) a natural language inference (NLI) component (Bowman et al., 2015) that models sentence-level semantics between a sentence containing a quantifier word and another sentence containing a percentage value, and (2) a rational speech act (RSA) component (Frank and Goodman, 2012) for pragmatic reasoning. Using these components, PRESQUE takes a sentence with a quantifier as input and outputs the corresponding percentage scope (further discussed in Section 2).

Ambiguity, as highlighted by Piantadosi et al. (2012), is beneficial for efficient communication via language. Since the percentage values of quantifiers are not universally defined, humans often need to infer the exact percentage value, which is not explicitly conveyed in the utterance (Horowitz et al., 2018). Furthermore, the interpretation of quantifier semantics can be influenced by linguistic and social cues (Bergen et al., 2016). The pragmatic theory proposed by Grice (1975) emphasizes the role of communicative goals in interpreting the semantic meaning of natural language expressions, simplifying the required semantic theories (Bergen

---

[1]Code: https://github.com/Nativeatom/PRESQUE

et al., 2016). Lastly, NLI models are also shown to struggle with ambiguous premises (Thukral et al., 2021), quantifier inference (Richardson et al., 2020; Joshi et al., 2020), and quantitative reasoning (Naik et al., 2018; Ravichander et al., 2019), making a direct literal interpretation of generalized quantifiers less reliable.

To address these challenges, PRESQUE employs RSA, a Bayesian framework that follows a Gricean approach for modeling communication by reasoning about agent states (Grice, 1975). PRESQUE incorporates a literal speaker role, based on a foundation model fine-tuned on NLI datasets, and a pragmatic listener role, computed using Bayesian rules, to reason between the quantifier space and the space of percentage values.

Existing datasets like HVD (Herbelot and Vecchi, 2015) and GQNLI (Cui et al., 2022) that investigate quantifier semantics either lack gold annotations of the percentage scopes for interpreting quantifier semantics, or are based on artificial setups using a small number of countable objects (Pezzelle et al., 2018b). Such fictional settings are not generalizable to broader and more complex real-world settings (e.g. describing concepts about a population using quantifiers). For a fair evaluation of the quantifier understanding capabilities acquired by foundation models through their pre-training, we should evaluate these models using text of similar style and content as the pre-training corpora. To address the aforementioned issues with current evaluation corpora for quantifier understanding, we crowd-source a dataset, QuRe (**Qu**antifier **Re**asoning), which contains sentences containing quantifiers paired with annotations for quantifier semantics in terms of percentage scopes. Additionally, we characterize the ease of making quantifier predictions for different sentences in QuRe.

Using PRESQUE to evaluate the quantifier reasoning ability of foundation models on QuRe, we observe a 20% span-based F1 boost over the literal listener baseline at all specificity levels (Section 5.2). Our experiments highlight the improved quantifier understanding of foundation models when approached from a pragmatic perspective rather than relying on direct interpretation using textual understanding frameworks like NLI. Although our framework does not explicitly model mathematical concepts, it is noteworthy that the mean strengths of quantifiers in foundation models, as revealed by PRESQUE, echo observations of quantifier hierarchies from previous works (Solt, 2016; Srivastava et al., 2018) that involve strong human priors, and findings from Pezzelle et al. (2018a), who associates quantifier usage with the counting behavior of human beings.

In summary, our contributions are two-fold: we develop PRESQUE based on pragmatic reasoning and NLI, and we crowd-source a dataset QuRe to support quantifier understanding investigation of foundation models. Our results on HVD and QuRe demonstrate that foundation models equipped with pragmatic reasoning (PRESQUE) can perform quantifier reasoning similar to humans.

## 2 Quantifier Semantics Understanding through RSA

We frame the task of quantifier understanding as the prediction of the percentage scope (e.g., 30%-50%) given a quantified sentence $\tilde{S}_q$ (e.g., *Some apples are red.*). Specifically, given an interval width $\beta$, we divide the percentage spectrum between 0 and 1 into evenly spaced intervals, denoted as $\mathcal{W}_\beta = \{p_i\}$ (e.g., $\mathcal{W}_{\beta=0.05} = \{0, 5\%, 10\%, ..., 95\%, 100\%\}$). The goal of a quantifier understanding model is to determine the percentage range in $\mathcal{W}_\beta$ where the associated predicate holds true (e.g., the proportion of red apples among all apples, 30%-50%).

To interpret quantifiers as percentage scopes, we develop PRESQUE, a framework that adopts the rational speech act (RSA) framework, with natural language inference (NLI) as the backbone for text understanding. The RSA framework consists of a speaker and a listener, where the listener infers the world state from the speaker's utterance by modeling the speaker's state of mind (Goodman and Frank, 2016). In PRESQUE, the world state corresponds to percentage values of predicates, while utterances correspond to quantifiers used with those predicates.

Given a premise $\tilde{p}$ (e.g., *Some apples are red.*) with quantifier $q$ (*some*) and a hypothesis $\tilde{h}$ (e.g., *30% apples are red.*) with a percentage value $p$ (*30%*), we use the entailment score between the premise and hypothesis, obtained from an NLI model, to define the literal listener $L_0$:

$$L_0(p|q) \propto \text{Entailment}(\tilde{p}, \tilde{h}) \qquad (1)$$

The pragmatic listener $L_1$, in the PRESQUE framework, interprets the semantics of the quanti-

fier word as:

$$L_1(p|q) \propto S_0(q|p)\mathbb{P}(p) \tag{2}$$

Here, $S_0$ represents a literal speaker that maps the semantics of percentage values to quantifier words. Practically, we model the speaker by swapping the premise and hypothesis in $L_0$:

$$S_0(q|p) \propto \text{Entailment}(\tilde{h}, \tilde{p}) \tag{3}$$

The prior $P(p)$ in Eq. 2 can be expanded as:

$$\mathbb{P}(p) = \sum_{q \in \mathcal{U}} \mathbb{P}(p|q)\mathbb{P}(q) \tag{4}$$

Here, $\mathbb{P}(p|q)$ is computed similarly to $L_0$, and $\mathbb{P}(q)$ represents the word frequency of $q$, obtained from the WORDFREQ dataset (Speer, 2022).

## 3 QuRe: Quantifier Reasoning Dataset

Existing datasets for quantifier understanding like HVD (Herbelot and Vecchi, 2015) are comprised of triples of the form ⟨concept, feature, quantifier⟩ (e.g. ⟨*shrimp, is_white, some*⟩) that denote how often a 'feature' is satisfied for a 'concept'. Notably, these datasets do not provide annotated percentage scopes that can be used to decipher the semantics of the quantifiers, i.e., how often (in numerical terms) the 'feature' is satisfied for the 'concept' in the real world, and the supporting documents (e.g. knowledge-bases in Wikipedia or any publicly available corpus) about the percentage scope of those triples are not easily accessible. Therefore, the judgments are based on subjective observation and experience (e.g. the proportion of white shrimps.) and are hence inaccurate. To address this shortcoming in available resources for quantifier understanding, we contribute a dataset, QuRe, for evaluating the quantifier understanding capabilities of language models. QuRe consists of sentences (from Wikipedia) containing percentage mentions annotated with the quantifiers.

Table 1 presents examples from QuRe. Of note, in addition to the quantifier annotation and percentage scopes, for each example in QuRe, we also provide specificity as additional metadata. Specificity measures the difficulty of deciphering the percentage scope of the quantifier from the sentence excluding the quantifier (i.e., if someone can deduce the percentage scope of a quantifier fully/partially from the sentence contents when the quantifier is absent; more details are provided later in Stage 4).

| WIKIPEDIA SENTENCE | [SPECIFICITY, EXPRESSION] QuRe SENTENCE |
|---|---|
| Squirrel Hill North's population is 75% White, 17% Asian, 4% Hispanic, and 3% Black. | **[Partial,** 0.04**]** Squirrel Hill North's population is 75% White, 17% Asian, few Hispanic, and 3% Black. |
| Coconut milk contains 5% to 20% fat, while coconut cream contains around 20% to 50% fat.. | **[Indeterminable,** $0.2 - 0.5$**]** Coconut milk contains 5% to 20% fat, while coconut cream contains moderate fat. |

Table 1: Examples of QuRe, with target percentage mention and the quantifier underlined. The headers of the QuRe also provide information about specificity and percentage expression generated. More examples are included in Appendix A.

The annotations in QuRe are obtained through a mix of crowd-sourcing (using Amazon Mechanical Turk) and assistance from gpt-3.5-turbo (OpenAI, 2022). We describe the annotation procedure in more detail below.

**Stage 1: Wikipedia sentences collection** We utilize the ⟨concept, feature, quantifier⟩ triples from the HVD dataset and convert them into sentences programmatically through templates (e.g. ⟨*shrimp, is_white, some*⟩ → 'some shrimps are white'). We then link these sentences to the most related Wikipedia entities[2] using an off-the-shelf library[3]. For example, the related Wikipedia entities for the running example would be *{Shrimp, Prawn, Indian prawn, etc.}*. In practice, we find this setting links to more diverse entities than simply linking the concepts. We then use regular expressions to extract around 5,000 candidate sentences containing percentage values from the Wikipedia pages of these entities. For example, *'Among the prawn species entering the field F. indicus constitute around 36%–43%.'* is one such sentence from the Wikipedia page of the entity *Indian prawn*, with the percentage mention underlined.

**Stage 2: Sentence Filtering** The candidate sentences are further filtered based on two criteria: (1) the length of the sentences should be between 10 and 40 tokens (space-tokenized), and (2) the percentage mentioned in the sentence should not indicate a comparative change (e.g., *'increased by 20%'*). To identify whether the sentence describes

---

[2] Each Wikipedia entity is the title of a Wikipedia article.
[3] https://pypi.org/project/wikipedia/

a comparative change, we used regular expressions. However, capturing all possible variations of describing such comparative changes through regular expressions is cumbersome. Hence we employ GPT-3.5-turbo to annotate sentences that contain comparative changes. To validate the efficacy of GPT-3.5-turbo, we manually annotate a held-out set of 50 sentences based on our aforementioned filtering criteria. On this held-out set GPT-3.5-turbo achieves 0.76 F1. More details on the annotation usage of GPT-3.5-turbo in this stage are included in Appendix H. The filtered sentences are then paired up with all percentage mentions in the sentence and manually validated by the authors. Around half of the percentage mentions were deemed inappropriate for the quantifier understanding task and removed. We include examples, metadata, and the instruction used in Appendix A and J.

**Stage 3: Percentage Expression Generation** In many Wikipedia sentences, the percentage value is surrounded by texts like *around*, *less than*, *more than*, etc. to denote a percentage scope rather than the individual percentage value or a percentage range. We use GPT-3.5-turbo to obtain those percentage scopes and the instruction is included in Appendix I. The variations that we capture in this stage to obtain the percentage scopes are mentioned in Table 2.

| Op. | Percentage Mention: Expression |
|-----|-------------------------------|
| None | 89%: 0.89 |
| $>$ | over 93%: $> 0.93$ |
| $>=$ | at least 45%: $>= 0.45$ |
| $<$ | less than 1%: $< 0.01$ |
| $<=$ | not exceeding 19%: $<= 0.19$ |
| $-$ | between 24% and 40%: $0.24 - 0.4$ |
| $\sim$ | about 98%: $\sim 0.98$ |

Table 2: Operators (**Op.**) in percentage expression generation and examples.

**Stage 4: Quantifier and Specificity Annotation** We design two human annotation tasks. The first task is rephrasing a sentence, $\tilde{S}_p$, with a target percentage mention (e.g. '*around 36%-43%*' of '*...the field F. indicus constitute around 36%–43%*' in the previous example) to $\tilde{S}_q$ with minimal semantic changes using a quantifier from $\mathcal{U}$ (e.g. *Among the prawn species entering the field, F. indicus constitute a large amount.*). This step ensures the seman-

tics of the quantifier used in $\tilde{S}_q$ is associated to the percentage scope in $\tilde{S}_p$.

In the second task, we measure specificity, or the difficulty of specifying the target percentage scope from $\tilde{S}_q$ without the quantifier $q$ (e.g. removing *a large amount* from the previous $\tilde{S}_q$). In our study, we discretize the specificity values into three distinct levels of difficulty: *full/partial/indeterminable*. *Full* means the target percentage scope can be fully determined by other contents in the sentence, like *One in ten* for *10%*; *partial* means the percentage scope can be narrowed but not determined by the contents (e.g. an incomplete percentage breakdown), and *indeterminable* means there is no information in the content of the sentence to determine the percentage scope. This task aims to gauge the extent of information that the context contributes to the determination of the quantifier's percentage scope. For example, the specificity of the previous $\tilde{S}_q$ about prawn would be *indeterminable* since the rest of the sentence after removing *a large amount* does not provide information to determine the percentage scope of *large*. But with additional contents like '*... constitute a large amount (around one-third).*', the specificity level would become *partially*. More examples are included in Appendix A.

We use majority voting (amongst three annotations) to choose the final annotated quantifier among all annotations for each sentence. The instruction used, examples, and example annotation interface are included in Appendix M. The set of quantifiers to select from is $\mathcal{U}$ = {*all, generally, most, usually, some, likely, few, little, occasionally, none, seldom, tiny, small, moderate, large*}, which largely comes from Srivastava et al. (2018), and is slightly extended based on preliminary annotator feedback. We leave the choice of nouns that are attached to adjective quantifiers (e.g. *amount* in *small amount*), like *small* and *large*, in sentences to the annotators.

**Statistics** We have collected 744 $\tilde{S}_q$ sentences, of which 47% and 17% contain no and one percentage mention respectively and others contain more than one percentage mention. The average sentence length is 26.3 tokens. Each sentence is annotated by 3 annotators. The Fleiss' Kappa for quantifier choices and specificity are 0.37 and 0.80, meaning fair agreement in quantifier choices and substantial agreement in specificity levels. The distribution of quantifiers in QuRe is shown in Fig-

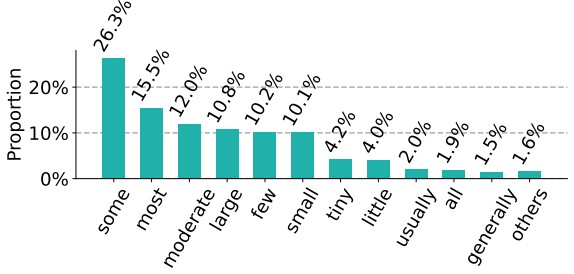

Figure 1: Distribution of quantifiers in QuRe. Quantifiers with less than 1% frequencies (*likely, seldom, occasionally, none*) are merged into *others*. *Some*, *most* and *moderate* are the most frequent quantifiers in QuRe.

ure 1, where *some* is used in over 25% of sentences, followed by *most*, *moderate*, *large* and *few*. The most popular quantifiers for different percentage scopes are shown in Figure 2, where *some* is preferred in over 30% of the cases with target percentage values lower than 40%, and *most* is selected in over 40% of the cases with target percentage value greater than 60%. Overall, 17% of the sentences have target percentages fully specified, 32% partially specified, and 50% are indeterminable. We include examples across different specificity levels in Appendix A.

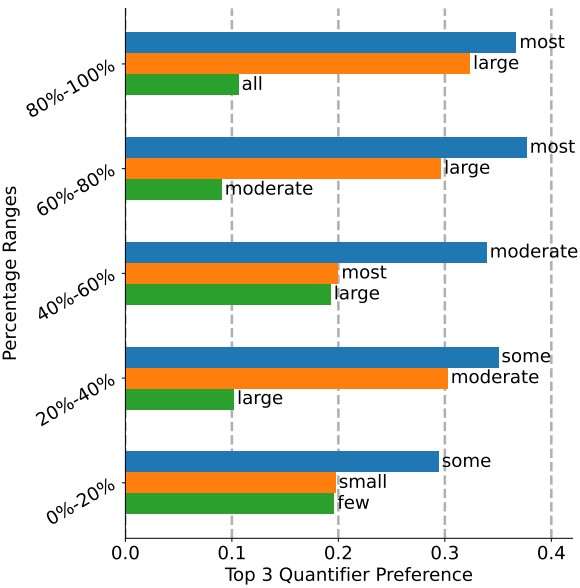

Figure 2: Quantifier preferences in difference percentage scopes, e.g., *most* is chosen around 35% of the times if the percentage mentioned lies in 60%-100%.

We also include the average strength of quantifiers under different grounding configurations in Table 3.[4] We can see that the mean strengths are

| Quantifier | $g = 0.01 \, w = 1$ | $g = 0.01 \, w = 4$ |
|---|---|---|
| all | $0.885 \pm 0.087$ | $0.892 \pm 0.085$ |
| generally | $0.730 \pm 0.205$ | $0.708 \pm 0.212$ |
| usually | $0.686 \pm 0.249$ | $0.674 \pm 0.242$ |
| most | $0.687 \pm 0.193$ | $0.693 \pm 0.195$ |
| large | $0.624 \pm 0.217$ | $0.628 \pm 0.223$ |
| likely | $0.473 \pm 0.287$ | $0.504 \pm 0.266$ |
| moderate | $0.369 \pm 0.154$ | $0.372 \pm 0.156$ |
| some | $0.225 \pm 0.185$ | $0.218 \pm 0.182$ |
| small | $0.183 \pm 0.184$ | $0.172 \pm 0.172$ |
| occasionally | $0.119 \pm 0.037$ | $0.124 \pm 0.037$ |
| seldom | $0.112 \pm 0.117$ | $0.093 \pm 0.106$ |
| little | $0.104 \pm 0.109$ | $0.117 \pm 0.135$ |
| few | $0.074 \pm 0.087$ | $0.081 \pm 0.098$ |
| tiny | $0.024 \pm 0.048$ | $0.031 \pm 0.046$ |
| none | $0.004 \pm 0.007$ | $0.004 \pm 0.007$ |

Table 3: Average strengths of quantifiers in all annotations of QuRe under different grounding configurations. The average strengths are stable with different window sizes.

stable among configurations, and show interesting hierarchies: *all* (0.88) is higher than *generally* (0.73), and *generally* is higher than *most* (0.69). These patterns closely align with previous manual strength assignments like Srivastava et al. (2018), and Testoni et al. (2019)'s quantifier collection from multimodal simulations. It also echoes Solt (2016)'s finding that the strength of *most* is higher than *more than half*.

## 4 Experimental setup

For the experiment in HVD, we compute $\mathrm{L}(p|q)$ for PRESQUE and $\mathrm{L}_0$ among different foundation models and compare them with human interpretations. In QuRe, with the target percentage given in Section 3, we generate the percentage scope that $\tilde{S}_q$ satisfies. All percentage choices are selected from $\mathcal{W}_\beta$, and experiments are run without training.

**Percentage Scope Generation** With specific granularity $g$ and window size $w$ for the operators, a percentage expression in Section 3 is converted into a golden scope $\{p_{\min}, p_{\max}\} \in \mathcal{W}_\beta$ ($p_{\min} \leq p_{\max}$). For example, if $\beta = 0.05$, $g = 0.01$ and $w = 2$, the golden scope of $\sim 0.59$ is $[0.55, 0.65]$. The full generation rules are in Appendix E.

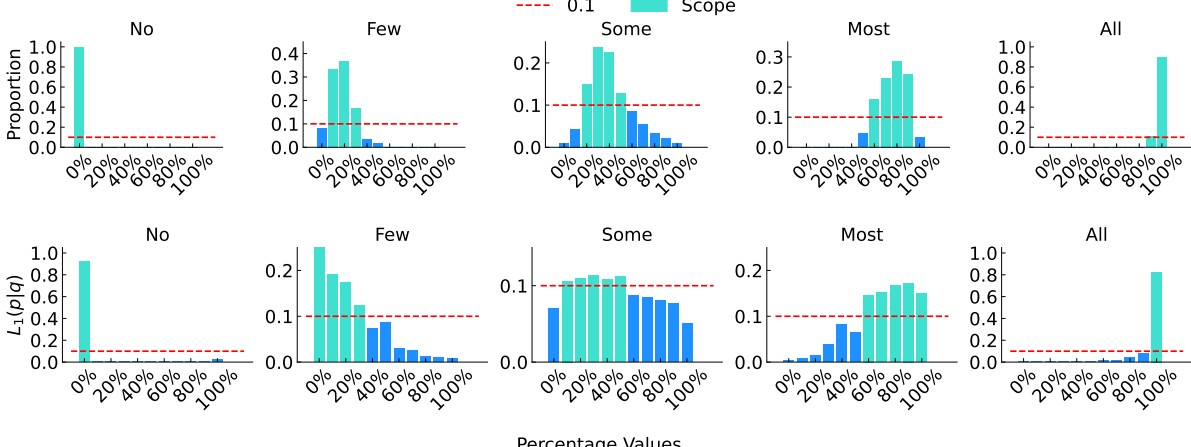

Figure 3: Human interpretations (upper) from 25 annotators and PRESQUE scores ($L_1$) from RoBERTa-large (bottom) of quantifier percentage scopes in HVD. The cyan bars indicate percentage values chosen by more than 10% of the annotators (red line) and, therefore could serve as approximate percentage scopes. For example, 10%-30% for *few* in human interpretations. The threshold is only used for illustration and not in experiments.

**Evaluation Metrics** For HVD, given a listener $L_M$ based on an NLI model M. $L_M(p|q)$ is computed by averaging the entailment scores over all $\tilde{S}_q$s for all $p$ values in $\mathcal{W}_\beta$ and normalize them to be a distribution. We can then compute the cross entropy between the human interpretation of quantifiers $\mathbb{P}_h$ from Section 5.1 and $L_M(p|q)$ to measure the similarity of quantifier interpretation between humans and M.

$$\text{CrossEntropy} = -\sum_{q\in\mathcal{U}}\sum_{p\in\mathcal{W}_\beta}\frac{\mathbb{P}_h(p|q)\log L_M(p|q)}{|\mathcal{U}|}$$

For $\tilde{S}_q$ in QuRe, we compute the following metrics,

$$\text{HIT@1} = \mathbb{I}[\arg\max_{p\in\mathcal{W}_\beta} L(p|q)\in s_{\text{golden}}]$$

$$\text{MRR} = \beta/(\text{B}_m\cdot\sum_{p'\in s_{\text{golden}}}\text{Rank}_{p'})$$

$$\text{CrossEntropy} = -\sum_{p'\in s_{\text{golden}}}\log\mathbb{P}(p'|q)$$

$$\text{where}\quad\mathbb{P}(p'|q)) = L(p'|q)/\sum_p L(p|q)$$

$$\text{where}\quad\text{B}_m = p_{\max}-p_{\min}+\beta,\quad p'\in\mathcal{W}_\beta$$

where $\mathbb{I}(\cdot)$ is an indicator, $s_{\text{golden}}$ is the gold scope $[p_{\min}, p_{\max}]$, and $\text{Rank}_{p'}$ is the rank of $p'$ in $\mathcal{W}_\beta$ by $L(p|q)$. HIT@1 measures whether the top inference percentage lies in the gold scope. MRR and cross entropy measure the average rank and confidence of the gold scope. We also compute the span-based F1 score between the gold scope and the primary scope (Section 5.2) of the top K

predictions (F1@K) under $\mathcal{W}_\beta$, which is used in question answering (Rajpurkar et al., 2016). The metrics are averaged over the entire dataset, and $L(p|q)$ is computed through Eq. 1 - Eq. 4.

## 5 Experiments and Results

We perform experiments to determine the percentage scope of quantifiers on two datasets: the HVD dataset, which includes predicates annotated with quantifiers but lacks percentage scope annotations, and the QuRe dataset, which provides annotations for both quantifiers and percentage scopes. As a baseline, we use the literal listener, $L_0$.

### 5.1 Human Interpretation of Quantifiers

To quantitatively assess the similarity between quantifier understanding between foundation models and humans, we first collect interpretations $\mathbb{P}_h(p|q)$ from human participants. For this, we employ 25 Mechanical Turk annotators who are tasked with providing the percentage scope of quantifiers. To guide them, we provide an instruction (see Appendix K for details) and present them with five questions. Each question requires the annotator to indicate the strength of a given quantifier word in $\mathcal{U}$ by providing either a percentage scope or a single percentage value in $\mathcal{W}_{\beta=0.1}$, without resorting to online searching. The distribution of the annotators' choices, as shown in Figure 3, reveals that humans interpret different percentage peaks for different quantifier words. The percentage scope of *few*, *some*, *most* indicated by the selection ratio of

more than 10% are larger than those of *no* and *all*. Meanwhile, the percentage scope of *some* is leaning to *few* rather than *most*, where *few* and *most* have little scope overlap.

## 5.2 NLI Model's Interpretation of Quantifiers

We evaluate the quantifier reasoning ability of the 'large' (or 'xxlarge') variants of AL-BERT (Lan et al., 2020), XLNet (Yang et al., 2019), BART (Lewis et al., 2020) and RoBERTa (Liu et al., 2019) that are fine-tuned on the NLI tasks (using Adversarial NLI (Nie et al., 2020), SNLI (Bowman et al., 2015) MNLI (Williams et al., 2018), and NLI-style FEVER (Nie et al., 2019) datasets).[5]

| BASE MODEL(#PARAM.) | CROSSENTROPY↓ | |
|---|---|---|
| | $L_0$ | PRESQUE |
| ALBERT (222M) | 1.76 | 1.48 |
| XLNet (361M) | **1.64** | 1.35 |
| BART (407M) | 1.89 | 1.32 |
| RoBERTa (355M) | 1.69 | **1.29** |

Table 4: Comparison of different NLI models in HVD with $L_0$ being the baseline of using NLI models for direct interpretation and PRESQUE is the pragmatic-based interpretation. PRESQUE is better than $L_0$ and RoBERTa-large has best cross entropy in PRESQUE.

The comparison of quantifier understanding using PRESQUE and $L_0$ is presented in Table 4. The results show that PRESQUE achieves lower cross entropies compared to $L_0$. Among the NLI models, RoBERTa performs the best within the PRESQUE framework, and therefore, it is chosen as the primary model for subsequent experiments. The $L(p|q)$ scores of PRESQUE from RoBERTa, which are used to represent the model's interpretation of the percentage scopes of different quantifiers, are displayed in the lower half of Figure 3. In general, different quantifier words exhibit distinct percentage distributions. Similar to Section 5.1, the scopes of *few*, *some*, and *most* can be approximated as *0% - 30%*, *10% - 50%*, and *60% - 100%*, respectively, with a cutoff criteria $L(p|q) \geq 0.1$. These ranges align closely with the scopes determined by human evaluation (upper half of Figure 3). Further, the $L(p|q)$ scores change in a smooth way as the percentages increase or decrease within the regions where $L(p|q) \geq 0.1$. This suggests that the model can understand percentage values quite well.

---

[5] In preliminary experiments, we found that foundation models without NLI fine-tuning performed worse on the quantifier prediction task.

| SENTENCE | SCOPE | PREF. |
|---|---|---|
| No ostriches are strange looking. | $L_0$: 0%-40% | 0.34 |
| | $L_1$: 0%-10% | **0.66** |
| Few tomatoes are green. | $L_0$: 0% | 0.12 |
| | $L_1$: 0%-30% | **0.88** |
| Some kites are made of plastic. | $L_0$: 80%-100% | 0.38 |
| | $L_1$: 10%-50% | **0.62** |
| Most owls live in forests. | $L_0$: 80%-100% | **0.66** |
| | $L_1$: 60%-100% | 0.34 |
| All gates are used for enclosing. | $L_0$: 60%-100% | 0.22 |
| | $L_1$: 70%-100% | **0.78** |

Table 5: Examples of percentage preferences between PRESQUE ($L_1$) and $L_0$ in HVD. The primary scope (Scope) is a scope with the maximum subarray sum of $L(p|q)$ within top K inference values, which stands for the most confident percentage scope of the model. The human preference (Pref.) is the ratio of scopes preferred by the human annotators. Green and blue represent preference to $L_0$ and PRESQUE, respectively.

Next, we compare the results of PRESQUE with that of a literal listener baseline, $L_0$ (Equation 1). As the percentage scope of a quantifier is measured by the consecutive percentage range among the top K-ranked percentage values, we compare the *consecutiveness* of $L_0$ and PRESQUE, which is measured by the proportion of sentences with the entire top K choices being able to constitute a single consecutive range. For example e.g. {*10%, 20%, 30%*} constitutes a consecutive range from 10% to 30% while {*10%, 30%, 50%*} does not. Consecutiveness is based on the assumption that consecutive ranking of percentage values indicates better quantifier understanding as the semantic meaning of quantifier words does not leap between disjoint percentage values. To enlarge the possible ranges, we start by K = 3 and include the results in Figure 4, where PRESQUE has higher consecutiveness than $L_0$, showing that PRESQUE has more consistent percentage inference behavior. Moreover, We select the primary scope by finding the consecutive scope (e.g. {*10%-30%*} and {*10%, 30%, 50%*} in the previous example) of the largest aggregated $L(p|q)$ among all consecutive scopes.

We additionally compare the primary scopes between $L_0$ and PRESQUE, through human preferences. For each quantifier word, we randomly sample 10 sentences where the top K inferences between $L_0$ and PRESQUE differ for the same $\tilde{S}_q$ with

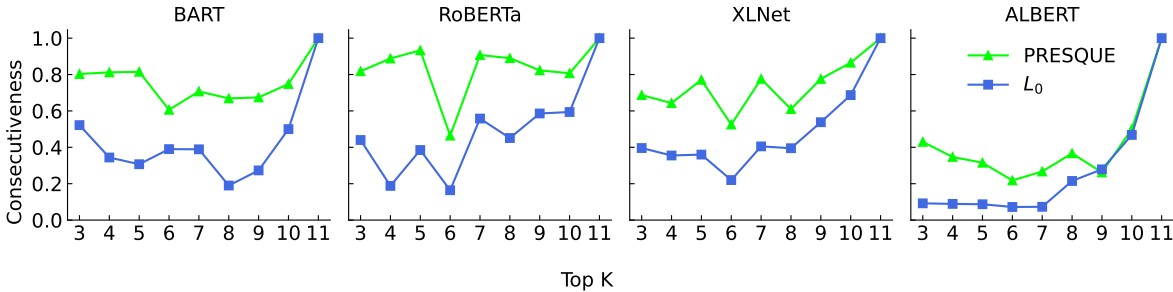

Figure 4: The ratio of top K percentage values from PRESQUE (lime) and $L_0$ (blue) that can form a single consecutive scope in HVD. PRESQUE has higher consecutiveness than $L_0$ among all models.

| SPECIFICITY | HIT@1↑ | | | MRR↑ | | | CROSSENTROPY↓ | | | F1@{1, 5}↑ | | |
|---|---|---|---|---|---|---|---|---|---|---|---|---|
| | Rnd. | $L_0$ | $L_1$ | Rnd. | $L_0$ | $L_1$ | Rnd. | $L_0$ | $L_1$ | Rnd. | $L_0$ | $L_1$ |
| Fully | 4.1 | 27.3 | **29.7** | 12.3 | 22.1 | **24.3** | 6.44 | **5.64** | 5.74 | 2.8/8.6 | 19.5/24.3 | **21.5/26.5** |
| Partial | 8.2 | 26.4 | **28.5** | 11.6 | 21.2 | **21.7** | 7.78 | **6.99** | 7.06 | 4.3/8.3 | 16.9/25.9 | **18.3/27.3** |
| Indeterminable | 9.7 | 21.4 | 21.4 | 12.5 | 18.1 | **22.7** | 7.76 | 7.20 | **6.69** | 5.3/10.1 | **14.9**/18.2 | 14.8/**25.6** |
| Total | 7.9 | 24.0 | **25.1** | 11.8 | 19.8 | **22.7** | 7.47 | 6.86 | **6.78** | 4.4/9.3 | 16.3/21.7 | **17.1/26.3** |

Table 6: Performance of PRESQUE ($L_1$) versus $L_0$ on QuRe using RoBERTa-large. Metrics are displayed on a 0-100 scale except for cross-entropy. *Rnd.* is a random baseline (averaged over 5 seeds) where $L(p|q)$ is sampled from $\mathcal{N}(0, 1)$ and normalized with softmax. The best results are bolded. The results show that the random baseline is worse than both $L_0$ and PRESQUE in most metrics. PRESQUE performs better than $L_0$ on almost all specificity levels and metrics.

K = 5. We then recruit 40 annotators from Amazon Mechanical Turk to select the more reasonable primary scope between $L_0$ and PRESQUE given $\tilde{S}_q$. We displayed the primary scopes for each $\tilde{S}_q$ in random order to avoid biases. Examples are included in Table 5 where PRESQUE generates smaller primary scopes for universal quantifiers like *No* and *All*, and larger primary scopes of other quantifiers which incorporate more vagueness. We leave the more general analysis in Appendix F.

Table 6 provides a comparison of the top percentage predictions with the gold scopes from PRESQUE and $L_0$ in QuRe. We observe that, in general, PRESQUE outperforms L0 in several aspects. Firstly, the topmost prediction value from PRESQUE appears more frequently within the gold scope, leading to a higher HIT@1 score. Additionally, the percentage values within the gold scope are ranked higher among the top predictions by PRESQUE, resulting in a higher Mean Reciprocal Rank (MRR). Furthermore, there is a larger overlap between the primary scopes of PRESQUE and the gold scopes, as indicated by a higher F1 score. Moreover, PRESQUE predicts better primary scopes on a distance-based metric designed to measure the distance between scopes, and we include the results in Appendix G. This finding aligns with the con-

clusion of Carcassi and Szymanik (2021), which suggests that listeners aim to minimize the distance from the speaker's observation when the communication success is measured based on distance.

**Qualitative Analysis.** Examining examples generated by PRESQUE, we make several interesting observations. For fully determinable contexts, such as "... only (2 out of the total 27) few school children were recognized..." with a gold scope of *5%-10%* (the true rate was 2 children out of 27 = 7%), PRESQUE provides a more accurate primary scope. In this case, PRESQUE predicted a scope of *0%-5%*, while $L_0$ predicted a scope of *0%*. For partially determinable contexts, such as "... calculating from less than few ... to 13%..." (indicating a partially determinable percentage scope of less than 13%), with a gold scope of *5%-10%*, PRESQUE often generates a broader scope than $L_0$. In this case, PRESQUE predicts *0%-15%*, which is more expansive than $L_0$'s prediction of *10%-15%*. For some indeterminable sentences like "... its alcohol content usually is very little." with a gold scope of *0%-5%*, PRESQUE provides a primary scope of *0%-5%*, while $L_0$ predicts a significantly distant scope of *60%-70%*. Appendix B provides a more comprehensive set of examples.

## 6 Conclusion

Generalized quantifiers are widely used for addressing satisfaction in natural language. However, the quantifier understanding abilities of foundation models are not well-studied or well-supported by the existing benchmarks based on factual context. In this work, we study quantifier understanding by proposing the `PRESQUE` framework that formulates the problem in a pragmatics reasoning perspective and the format of NLI. And we collect a dataset `QuRe` that includes human annotated quantifiers and percentage mentions for Wikipedia sentences. From the experimental results on the HVD dataset and our collected `QuRe` dataset, the `PRESQUE` achieves better performance compared to the direct interpretation of quantifier semantics by a literal listener model.

## Acknowledgments

The authors would like to thank the anonymous reviewers for their suggestions and feedback on the work. This work was supported in part by NSF grant DRL2112635. The views contained in this article are those of the authors and not of the funding agency. We also thank Daniel Fried for useful discussion and suggestions to this work.

## Limitations

In this work, we investigate the quantifier understanding abilities of several foundation models and collect a dataset `QuRe` that we expect will substantially benefit research on quantifier semantics. However, despite the value of our dataset and the promising results from the `PRESQUE` framework, our analysis and findings have some notable limitations. First, we note that our study and dataset still focus on a small part of the generalized quantifiers and likely do not cover the entire spectrum of quantifier semantics. Second, the sentences in our dataset all come from Wikipedia. Consequently, the performance of `PRESQUE` and the generalizability of our findings to other domains or languages remains an open question. Finally, assigning precise percentage scopes to quantifiers can be a challenging or even impossible task, since quantifier semantics are complex and depend on many factors beyond those analyzed here. In particular, these may subjectively depend on the domain, an annotator's background of knowledge or culture, comfort with the mathematics of percentages, and Bayesian vs Frequentist interpretations of percentage numbers, among many other factors. Thus, ambiguities and subjectivity are natural when determining a quantifier's scope. Our dataset and analysis largely skirt many of these complex issues.

## Ethics and Broader Impact

We employ crowdsourcing through Amazon Mechanical Turk for (a) certain annotations of our dataset, `QuRe`, (b) understanding human interpretation of quantifier semantics, and (c) human evaluation of `PRESQUE` and baselines. In all our crowdsourcing tasks we do not collect any personally identifiable information about the turkers and all the turkers were paid above minimum wage, which is included in Appendix D. We released our crowdsourcing tasks to turkers in the USA and constrained the approval rate of the turkers to be above 98% to ensure good-faith turkers.

Besides, the prevalence of quantifiers in naturally occurring corpora would inherit the generation behavior of models. `PRESQUE`, as one step towards revealing the quantifier understanding ability of foundation models, could be helpful in more accurately interpreting the meaning in model-generated text. It could also support automatic pipelines for knowledge-intensive reasoning that include quantifications, or logical reasoning in natural language.

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

# A   Example Metadata of `QuRe`

To investigate the topic coverage of `QuRe` sentences, we use GPT-3.5-turbo to generate 3 topics for each sentence, using instruction in Appendix J. The most frequent topics are listed in Figure 5, where nearly 10% sentences are about *statistics*, followed by *animal*, *percentage* and *demographics*.

# B   `PRESQUE` Examples in `QuRe`

We include several examples in Table 8 where $\tilde{S}_q$ as well as the specificity level, and the $\tilde{S}_p$ as well as the golden percentage scope are located on the upper and lower half of each block. We can see that

| [Wiki entity] Original Sentences | [Specificity, Expression] QuRe Sentences | Topics |
|---|---|---|
| **[Human]** Most humans (61%) live in Asia; the remainder live in the Americas (14%), Africa (14%), Europe (11%), and Oceania (0.5%).Within the last century, humans have explored challenging environments such as Antarctica, the deep sea, and outer space. | **[Fully,** 0.11] Most humans (61%) live in Asia; the remainder live in the Americas (14%), Africa (14%), some Europe, and Oceania (0.5%).Within the last century, humans have explored challenging environments such as Antarctica, the deep sea, and outer space. | population continents exploration |
| **[The Jungle Book (2016 film)]** The Jungle Book was shown across 4,028 theaters of which 3,100 theaters (75%) were in 3D, including 376 IMAX screens, 463 premium large format screens, and 145 D-Box locations. | **[Fully,** 0.75] The Jungle Book was shown across 4,028 theaters of which most (3,100) theaters were in 3D, including 376 IMAX screens, 463 premium large format screens, and 145 D-Box locations. | theaters movie release 3D technology |
| **[Electric car use by country]** The EV market share of total new and used cars first registered during 2018 was 2.8% based on 5,557 out of a total of 198,600 first registered cars.7,542 vehicles were registered in this country over 2019. | **[Fully,** 0.028] The EV market share of total new and used cars first registered during 2018 was small based on 5,557 out of a total of 198,600 first registered cars. 7,542 vehicles were registered in this country in 2019. | electric vehicles market share registration numbers |
| **[List of blade materials]** Prior to 2002, INFI contained 0.5% Carbon, 0.74% Nitrogen, about 1% Cobalt, and about 0.1% Nickel. | **[Partially,** 0.005] Prior to 2002, INFI contained tiny levels of Carbon, 0.74% Nitrogen, about 1% Cobalt, and about 0.1% Nickel. | chemical composition INFI elements |
| **[Housing in the United Kingdom]** British dwellings had the oldest age profile in the EU with over 60% being built before 1960, and with only just over 10% being built between 1991-2010. | **[Partially,** $> 0.1$] British dwellings had the oldest age profile in the EU with over 60% being built before 1960, and with some being built between 1991–2010. | age housing statistics construction date |
| **[Ice cream]** Gelato typically contains 7-8% fat, less than ice cream's minimum of 10%. | **[Partially,** $>= 0.1$] Gelato typically contains 7–8% fat, less than the moderate amount found in ice cream. | food comparison fat percentage |
| **[Tobacco]** A study published in Morbidity and Mortality Weekly Report found that in 2019 approximately one in four youths (23.0%) in the U.S. had used a tobacco product during the past 30 days. | **[Partially,** 0.23] A study published in Morbidity and Mortality Weekly Report found that in 2019, some (one in four) youths in the U.S. had used a tobacco product during the past 30 days. | youth tobacco use scientific study |
| **[Polish cuisine]** It is typically made from rye bread, usually known as black bread, and is not classified as an alcoholic beverage in Poland, as its alcohol content usually ranges from 0% to 2%. | **[Indeterminable,** $0-0.02$] It is typically made from rye bread, usually known as black bread, and is not classified as an alcoholic beverage in Poland, as its alcohol content usually is very little. | food beverage alcohol content |
| **[List of blade materials]** In order for a steel to be considered stainless it must have a Chromium content of at least 10.5%. | **[Indeterminable,** $>= 0.105$] In order for a steel to be considered stainless it must have some Chromium content. | steel metallurgy composition |
| **[British military bands]** The average age of the 304 drummers at Waterloo was 25, with about 10% being boys under 16. | **[Indeterminable,** $\sim 0.1$] The average age of the 304 drummers at Waterloo was 25, with some being boys under 16. | age music statistics |

Table 7: Example data of QuRe, with target percentage mention and quantification underlined. The header marks either the Wikipedia entity where the sentence is extracted or the specificity and the generated percentage scopes. For example, for the Jungle Book sentence, the percentage scope 75% can be fully specified by the proportion of 3100 over 4028 theatres, while for the sentence about Gelato, the content before the percentage mention indicates that the fat content of ice cream is higher than 7-8%, but cannot provide more information to further narrow down the scope, and therefore the specificity is *partially*.

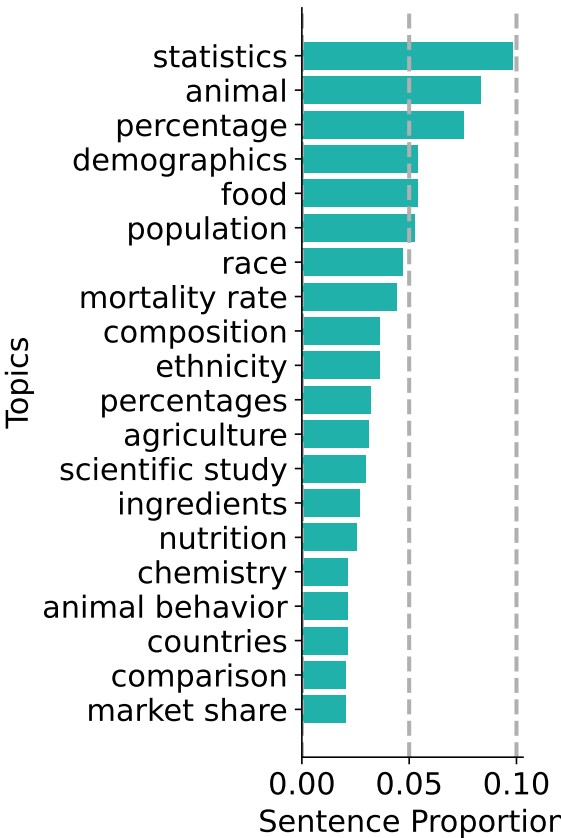

Figure 5: Top 20 sentence topic statistics in QuRe. The most frequent sentence topics, where nearly 10% sentences include the topic *statistics* and 8% sentences include *animal*. Topics like *demographics*, *food* and *population* also cover more than 5% sentences in QuRe.

PRESQUE provides more accurate primary scopes for fully determinable sentences.

## C  Discussion

The semantic understanding of a pragmatic listener is proportional to the product of two entailment-based probabilities, where the premise and hypothesis are flipped with respect to each other (i.e., the premise used for $S_0(q|p)$ is the hypothesis for calculating $P(p|q)$ in the prior (Eq. 4) To arrive at an intuitive understanding of why considering the flipped premise-hypothesis pairs, we analyze the sensitivity of NLI models (specifically, RoBERTa fine-tuned NLI model) towards percentage values (quantifier inference) and quantifier words (percentage inference) in premises. The distribution of average entailment scores over all the premises is shown in Figure 6. The upper part of the figure shows the result of percentage inference and the lower part shows the result of quantifier inference,

with two thresholds, 0.1 and 0.5. We can observe that the NLI model is more sensitive to the percentage values in the premise than quantifiers. The entailment scores of percentage inference is relatively low, which is led by high neutral scores, making it challenging to identify the percentage scope for each quantifier. For example, *few*, *some* and *most* don't have any percentage values that exceed even the lower threshold. The lower half of Figure 6, however, demonstrates more interpretable entailment distribution, where the percentage scopes of *few*, *some* and *most* can be interpreted as *0%-20%*, *0%-90%* and *60%-100%* by the higher threshold. In short, NLI models are more sensitive to interpreting accurate numerical premises, which has also been observed that NLI performs better with accurate premises (Thukral et al., 2021; Richardson et al., 2019) where premises with quantifiers are less accurate than premises with percentage values.

## D  Annotator Recruitment

We use Amazon Mechanical Turk as the crowd-sourcing platform for different annotation aspects of QuRe as described in previous paragraphs. To finalize our pool of turkers, we released a qualification task to test basic understanding of quantifier semantics. The annotators are selected to base on the United States, completed more than 1000 HITs with more than 98% approval rate. For annotator that participates the final QuRe collection in Section 3, they can make at most one annotation mistake in sentence rephrasing of the qualification task. The qualification task had a pass rate of 38% and we recruited 18 turkers for the main annotation tasks of QuRe. Annotators are paid about $ 7.30/hr on average. Besides, annotators recruited for human interpretation of quantifiers are paid about $ 9/hr on average. And the annotators recruited for human preference of percentage scopes are paid about $ 9.60/hr on average.

## E  Percentage Scope Generation Details

With granularity $g$, window size $w$, the grounded percentage scope can be determined in Table 10.

For HVD, $\beta$ is set to be 0.1. And in experiments on QuRe, $\beta$, $w$, $g$ are set to be 0.05, 2, 0.01 respectively unless specified.

## F  Human Preference of HVD Examples

Figure 7 shows PRESQUE is in general preferred over $L_0$ by the annotators, while the preferences

| [GS.] SENTENCE$_Q$ / [SPC.] SENTENCE$_P$ | PRIMARY SCOPE | MRR | F1@5 | CE |
|---|---|---|---|---|
| **[F]** In 57 separate fights, one loss was observed to Neope goschkevitschii, giving V. mandarinia a large winning rate. | $L_0$: 5%-20% | 0.11 | 0.00 | 7.67 |
| **[95%-100%]** In 57 separate fights, one loss was observed to Neope goschkevitschii, giving V. mandarinia a win rate of 98.3%. | $L_1$: 85%-100% | **0.67** | **0.67** | **3.52** |
| **[F]** In the 2017 Dutch study, only (2 out of the total 27) few school children recognized that the website was a hoax. | $L_0$: 0% | 0.08 | 0.00 | 7.79 |
| **[5%-10%]** In the 2017 Dutch study only 2 out of the total 27 school children (7%) recognized that the website was a hoax. | $L_1$: 0%-5% | **0.11** | **0.50** | **6.36** |
| **[P]** From 4 locations in different parts of Europe, a large number had clutch size of 2, 41% had size of 3, clutches of 1 and 4 each constituted about 8%. | $L_0$: 30%-40% | 0.22 | 0.40 | 6.29 |
| **[40%-45%]** From 4 locations in different parts of Europe, 43% had clutch size of 2, 41% had size of 3, clutches of 1 and 4 each constituted about 8%. | $L_1$: 30%-45% | **0.33** | **0.67** | **4.92** |
| **[P]** The empirical occurrence of regenerated claws in fishery harvests is low, with studies on stone crabs calculating from less than few (Davis et al., 1978), to 13% (Florida Fish and Wildlife Conservation Commission, 2011). | $L_0$: 10%-15% | 0.17 | 0.50 | 7.79 |
| **[5%-10%]** The empirical occurrence of regenerated claws in fishery harvests is low, with studies on stone crabs calculating from less than 10% (Davis et al., 1978), to 13% (Florida Fish and Wildlife Conservation Commission, 2011). | $L_1$: 0%-15% | **0.50** | **0.67** | **4.40** |
| **[I]** It is typically made from rye bread, usually known as black bread, and is not classified as an alcoholic beverage in Poland, as its alcohol content usually is very little. | $L_0$: 60%-70% | 0.06 | 0.00 | 6.97 |
| **[0-5%]** It is typically made from rye bread, usually known as black bread, and is not classified as an alcoholic beverage in Poland, as its alcohol content usually ranges from 0% to 2%. | $L_1$: 0%-5% | **0.33** | **1.00** | **4.16** |
| **[I]** Chlamydospore germination requires 30 to 52 hours, with a moderate germination success rate. Spore production is highest at midday, relative to temperature increase and relative humidity decrease. | $L_0$: 30%-35% | 0.13 | 0.50 | 18.85 |
| **[30%-55%]** Chlamydospore germination requires 30 to 52 hours, with a germination success rate of 32 to 54%. Spore production is highest at midday, relative to temperature increase and relative humidity decrease. | $L_1$: 40%-50% | **0.22** | **0.67** | **16.17** |

Table 8: Examples of PRESQUE ($L_1$) versus $L_0$. The sentences are paired with percentages and the corresponding sentence with quantifiers, with the target percentage and quantification phrase underlined. The headings mark either the gold scope (**GS**) or the specificity levels (**SPC.**) with **[F/P/I]** being fully/partially/indeterminable respectively. **CE** stands for cross-entropy. Bolded figures are better results. Predictions are collected from $\mathcal{W}_{\beta=0.05}$. $L_1$ achieves better MRR and cross entropy then $L_0$ among different sentence inferring categories.

| CONCEPT | FEATURE | ANNOTATIONS | SENTENCE BASED ON TEMPLATE |
|---|---|---|---|
| rock | has_minerals | all, all, most | All rocks have minerals. |
| van | has_sliding_doors | most, most, most | Most vans have sliding doors. |
| sandpaper | has_fine_sand_covering_it | some, some, all | Some sandpapers have fine sand covering it. |
| banana | is_round | no, no, no | No bananas are round. |
| tricycle | used_for_transportation | all, few, few | Few tricycles are used for transportation. |

Table 9: Sample ⟨concept, feature⟩, human annotations for the quantifiers, and the corresponding HVD sentences that serve as $\tilde{S}_q$ using the majority quantifier annotation.

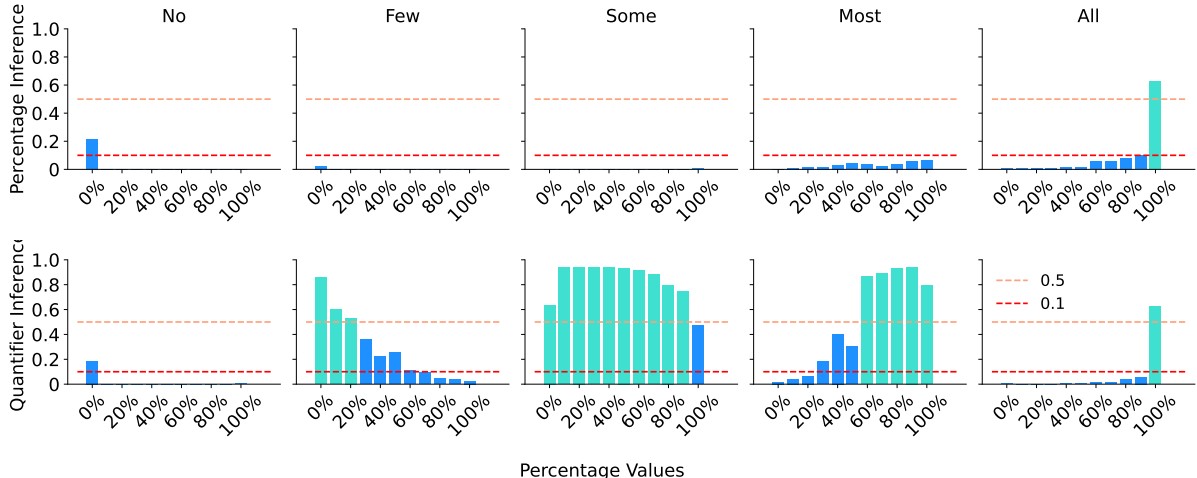

Figure 6: Comparison of the NLI model conducting percentage inference (Entailment($\tilde{S}_q, \tilde{S}_p$), upper) and quantifier inference (Entailment($\tilde{S}_p, \tilde{S}_q$), lower) in HVD. The NLI model is more sensitive in quantifier inference in general. Cyan bars indicate values higher than the upper threshold (0.5). Note that the bar values do not stand for probabilities and do not sum up to 1.

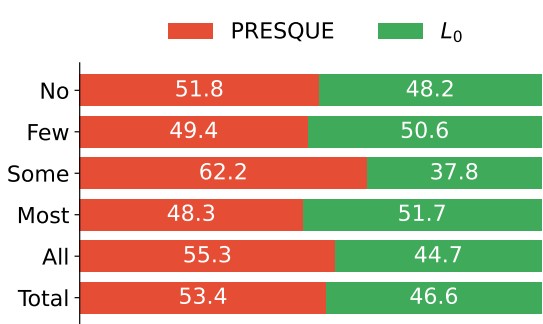

Figure 7: Listener preference from humans for HVD examples. The inference from PRESQUE is in general, preferred by humans than $L_0$, while the preference of each quantifier differs (PRESQUE is more preferred for *No, Some* and *All*).

| EXPRESSION | PERCENTAGE SCOPE |
|---|---|
| $p$ | $p$ |
| $> p$ | $(p, p + w \cdot g]$ |
| $>= p$ | $[p, p + w \cdot g]$ |
| $< p$ | $[p - w \cdot g, p)$ |
| $<= p$ | $[p - w \cdot g, p]$ |
| $p_1 - p_2$ | $[p_1, p_2]$ |
| $\sim p$ | $[p - w \cdot g, p + w \cdot g]$ |

Table 10: Percentage scope grounding rules with granularity $g$, window size for approximation $w$. And $[p_{\min}, p_{\max}]$ is the smallest scope in $\mathcal{W}_\beta$ that includes the above scope. The scope would be cut off at 0 and 1.

## H Instruction for Sentence Filtering

In this task, you will determine whether a given sentence that has one or more quantifier values mentioned can have those quantifier values replaced by a natural language quantifier like 'some', 'most' or 'generally'.

Example sentences that meet the criteria are like 'It consists of about 80% water, soluble minerals (nearly 3% with half of the potassium) and polyphenols.' and sentences that don't meet the requirement are like '180.1 million were rides on SEPTA's 'city transit' network. Ridership had decreased 13% from 2014 to 2019 due to many factors.' where the percentage value represents incremental percentage changes or comparisons (e.g. 'drop by 50%' or '20% higher', 'X% better') instead of absolute percentages.

Do you think the following sentence meets the requirement? Answer in Yes or No:

may differ for different quantifiers. The primary scopes of PRESQUE for *no*, *some* and *all* are more preferred than $L_0$ by the annotators. *Some* and *all* have $p < 0.05$ in chi-squared test.

## G Distance-based Scope Evaluation

To measure the primary scope of $L(p|q)$ and the gold scope $[p_{\min}, p_{\max}]$ in distance-based metrics. We compute the minimal scope distance (MSD) over the top K predictions (MSD@K). Specifically,

$$\text{MSD@K} = \sum_{p' \in \text{TopK } p} \frac{\mathbb{I}[p' \notin s_{\text{golden}}]}{\text{B}_m} \min(p_{\min} - p', p' - p_{\max})$$

$$\text{where} \quad \text{B}_m = p_{\max} - p_{\min} + \beta, \quad p' \in \mathcal{W}_\beta$$

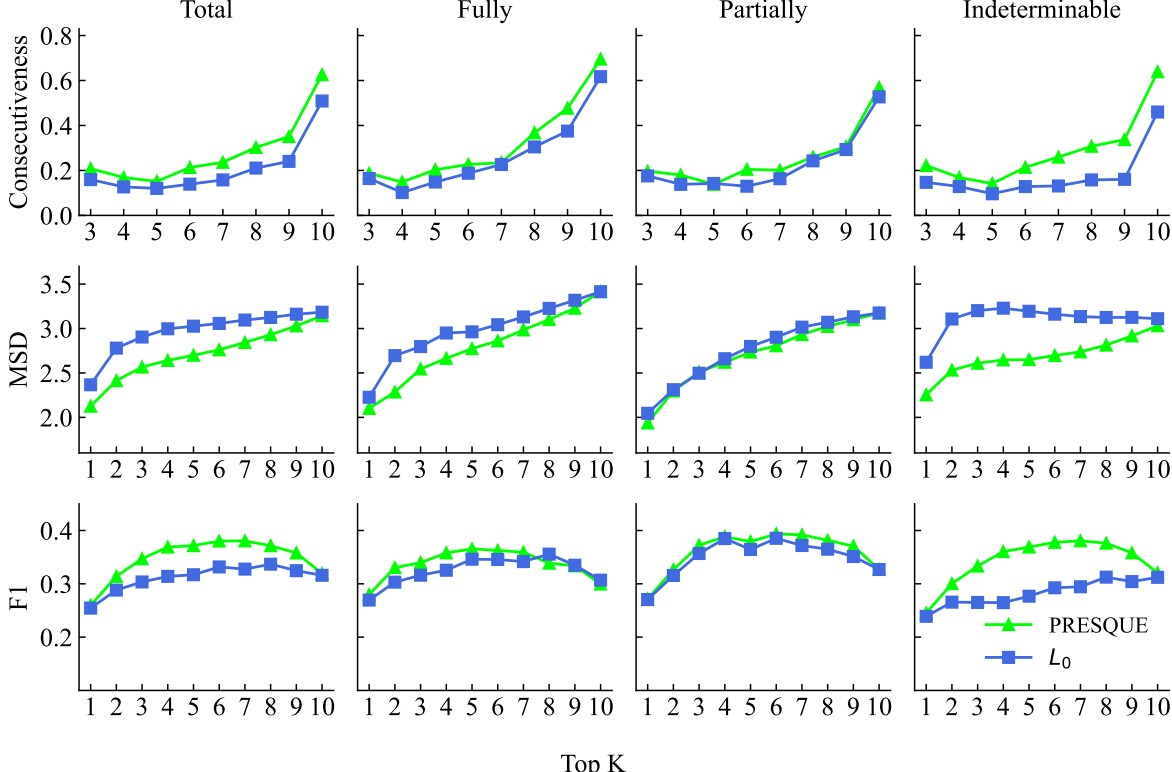

Figure 8: Consecutiveness (↑), MSD (↓) and F1 (↑) of PRESQUE (lime) and $L_0$ (blue) on `QuRe` with $\beta = 0.1$ for better illustration. PRESQUE has higher consecutiveness, lower MSD and higher F1 than $L_0$ across all specificity levels.

## I Instruction for Percentage Scope Generation

The instruction for percentage scope generation is shown below.

> In this task, you will give a sentence and a quantifier expression in the sentence, and you need to convert that quantifier expression into a mathematical expression. For example, for the sentence 'About 30% of homes are owned outright by their occupants. [30%]', you are given 30% in the bracket, and the corresponding mathematical expression is ~0.3, where ~ means approximate. Similarly, all the other mathematical operations supported include > meaning 'more than' (e.g. 'more than 80%' would be > 0.8), < meaning 'less than', '-' meaning range (e.g. '20% to 50%' would be 0.2-0.5) and null meaning exact (e.g. 'takes up 20%' would be 0.2). Answering the expression itself is enough, don't repeat the sentence or use additional English words other than the operations. Also, try to avoid using '<' and '>' if you can formulate the range by using '-'. Now, please do the same conversion for the following sentence:

## J Instruction for Sentence Topic Generation

> Please use three or four labels to categorize a given sentence (starts with "sentence"), including the topics of the contents, split with semicolons.
> For example,
> Sentence: In fact, a 2006 survey found that trapping as a solution to beaver problems had a 79% failure rate within two years due to resettlement by new beavers. Labels: scientific study; animal; rate
> Sentence: Among individual countries, the proportion of urban residents living in slum areas in 2009 was highest in the Central African Republic (95.9%), Chad (89.3%), Niger (81.7%), and Mozambique (80.5%). The distribution of slums within a city varies throughout the world. Labels: population; ranking; countries.
> Now, please label the following sentence:

## K Instruction for Human Evaluation

The instruction for collecting human perception of quantifier words in Section 5.1 is displayed as

This form contains several natural language quantifiers. The users are expected to pick one/two numerical percentages from the provided list of percentages such that most accurately bound the range of the given quantifier to the best of his/her knowledge and online searching is not encouraged.

Users can use different (real or imaginative) statements as examples to help estimate the range, such as 'No water comes from the sky.' and 'Most sea cucumbers are scavengers.'.

The users can select no more than 2 options to mark the lower and upper bound of the range, if they believe only one percentage would apply, they can select only 1 option.

An example of 'All' stands for' with a statement is 'All sugars are white.' The users are expected to select '100%' or a range (based on the user's understanding) from all provided percentages as the range for 'All', and the paraphrase becomes 'A% to B% sugars are white.' (A and B are the selections and can be the same) which becomes the most appropriate paraphrase of 'All sugars are white'. Note that the statement itself does not necessarily involve factuality (in fact, sugars can have various colors).

---

The instruction for collecting listener preference of $L_0$ and $L_1$ in Section 5.2 is displayed as

---

This form contains several statements (e.g. sugars are white) with natural language quantifiers (e.g. all). The users are expected to pick the more appropriate percentage range from the provided two options such that accurately bound the range of the given quantifier to the best of his/her knowledge and online searching is not encouraged.

An example statement with quantifier is 'All sugars are white.', and two example options are '0%-30%' and '90%-100%'.

In this example, the users are expected to select '90%-100%', which results in that '90%-100% sugars are white.' better describes 'All sugars are white.'. Note that the statement itself does not necessarily involve factuality (in fact, sugars can have various colors).

## L  Quantifier Understanding of GPT-3.5-turbo

Although we mainly focus on NLI models to develop PRESQUE, we also test the performance of QuRe on GPT-3.5-turbo using the following instruction.

---

In this task, you are given a sentence (starts with 'Sentence:') containing a predicate with a quantifier, and you need to provide a percentage scope that the predicate satisfies.

For example, if you are given "Sentence: Some apples are red." for the quantifier 'some', and you believe 37%-42% apples are red. Then the percentage scope for "some apples are red" would be 37%-42%.

The scope you can choose should be rounded in the granularity of 5 %. In the previous 'apples are red' example, your answer will be "35%-45%". Not that the percentage value cannot exceed 0% and 100%. You can also select one single percentage value for the scope.

Please provide a percentage scope for "some" in the following sentence.

Sentence:

---

In the example instruction shown above, the quantifier *some* would be replaced by the target quantifier that appeared in the sentence attached to the instruction. For example, for sentence *Adult clams can get most of their nutrients from the algae and the rest from filter feeding.* (gold scope 65%-100%), the output of GPT-turbo-3.5 for quantifier *most* is *60%-80%*.

Overall, GPT-3.5-turbo achieves 0.28 F1 score of the quantifier understanding task on QuRe, under the same configuration in Appendix E, which is slightly higher than the F1@5 performance of PRESQUE. However, we are aware that text-to-text models like GPT-3.5-turbo still suffer from hallucination and the output is unstable due to temperature-based sampling. Meanwhile, the PRESQUE in this work is agnostic to the backbone model choices and can be applied to any models that score the entailment relation between sentences.

## M  Annotation Task Interface

The instruction for the qualification task in collecting QuRe is included in Figure 9, and Figure 10 shows the example tasks annotators need to complete.

**Overview**

The objective of this task is to associate numerical percentages to quantifiers (e.g. 'always'). Given a sentence in English containing one or more percentage values, replace a given percentage value/range with a suitable English language quantifier so that the meaning of sentences remains as unchanged as possible. The suggested list of quantifiers and steps for the task are provided below. Please read the instructions carefully before proceeding.

**Instruction:**

1. Given the original English sentence and a numerical percentage, please write one quantified sentences has the closest meaning to the original sentence by replacing the phrase with given numerical percentage with a quantifier word.
   You need to use one of the following quantifier words:

   all, generally, most, usually, some, likely, few, little, occasionally, none, seldom, tiny, small, moderate, large

   Fill **NotApplicable** if none of the quantifiers can be applied, but try your best to fit one of them.

   Use External Reference Link for additional context to help determine the quantifier word for the given sentence.
2. Also, please conduct necessary paraphrasing to make the quantified sentence fluent with minimal changes to the sentence meaning (removing some words in the original sentence is also allowed). For example, the quantifier word is not likely to appear isolately in parentheses.
3. Please select one from the following options about how much the replaced percentage value can be specified from the non-quantifier context in the quantified sentence (making best guess if not sure).

   **Fully**: the exact percentage value can be specified from the context by some figures mentioned in the context. [Hide Example]

   "*[Quantifier] (3 out of 5) cars are broken on the street.*" (The percentage value of [Quantifier] can be specified from 3 out of 5)

   **Partially**: the exact percentage value can not be determined, but the scope can be specified from other figures or percentages in the context. [Hide Example]

   "*[Quantifier] flowers are red, 30% flowers are blue, these are the major colors.*". (the [Quantifier] refers to percentage value between 0% and 70%)

   **Indeterminable**: the context does not provide enough information to either determine the scope of the replaced percentage value. [Hide Example]

   "*[Quantifier] birds can fly.*"(no enough information provided for the [Quantifier])

4. Optional: after finish the task, feel free to provide useful feedback, any confusions to help improve it in the feedback box. Thanks.

Important Information before proceed

1. The total number of sentences to finish is **10**.
2. Your answer will be evaluated by the following criteria (special attention to **Fluency** to receive the reward),
   - **Completeness**: Complete all tasks (except optional feedback) to receive the reward.
   - **Correctness**: The qualifications are respondes to the phrase about the given percentage value or ranges and uses the GIVEN quantifier.
   - **Sentence Fullness**: Fill in full quantified sentences instead of merely the quantification, and remove the corresponding percentage value.
   - **Reasonableness**: Whether the quantification uses a reasonable quantifier word to replace the percentage value.
   - **Fluency**: Use necessary edits to make the final sentences fluent in English.

[Hide Examples]

**Examples:**

Original sentence: About 30% of homes are owned outright by their occupants. [30%]
(Correct) Quantified sentence: **Some** homes are owned outright by their occupants.
(Incorrect) Quantified sentence: **Some** (about 30%) homes are owned outright by their occupants. ( Sentence Fullness ✖ )
Specificity: *Indeterminable*

Original sentence: The milk of these species consist of up to 60% fat, allowing the young to grow fairly quickly. [60%]
(Correct) Quantified sentence: **Most** substances in the milk of these species are fat, allowing the young to grow fairly quickly.
(Incorrect) Quantified sentence: The milk of these species consist of up to **most** fat, allowing the young to grow fairly quickly. ( Fluency ✖ )
(Incorrect) Quantified sentence: **None** (60%) substances in the milk of these species are fat, allowing the young to grow fairly quickly. ( Reasonableness ✖, Sentence Fullness ✖ )
Specificity: *Indeterminable*

Figure 9: Instruction for the annotation task.

Those who violate the law face a fine of up to CHF 10,000.In September 2018, a ban on face-covering veils was approved with a 67% vote in favour in the canton of St. Gallen. [67%]

☐ Show External Reference Link (Optional)

Quantified sentence:

Specificity
○ Fully  ○ Partially  ○ Indeterminable

The mortality rate among untreated bite victims is nearly 100%. [100%]

☑ Show External Reference Link (Optional)

link (recommend to open in a new tab)

Quantified sentence:

Specificity
○ Fully  ○ Partially  ○ Indeterminable

Feedback (Optional)

Submit

Figure 10: Interface of the example annotation task. Each sentence comes with a target percentage in the bracket at the end of the sentence that directs the target percentage mentioned in the sentence (e.g. *100%* for *nearly 100%* in the second sentence). If there are multiple target percentage mentions that share the percentage value, a positional indicator would be attached. Besides, an optional reference link is provided by checking the box.