# OpenReview forum: "Pragmatic Reasoning Unlocks Quantifier Semantics for Foundation Models"
_EMNLP/2023/Conference — EMNLP 2023 Main_

### Official Review · Reviewer_PLvc · 2023-08-03

**Soundness:** 4

**Excitement:**

4: Strong: This paper deepens the understanding of some phenomenon or lowers the barriers to an existing research direction.

**Paper Topic And Main Contributions:**

- This paper presents QuRe, a dataset for converting generalized quantifiers into percentages based on crowdsourcing, and proposes the PRESQUE framework, which combines an RSA framework based on NLI and Gricean theory.
- PRESQUE shows an improvement of about 20% in predicting the percentage scopes of generalized quantifiers.

**Questions For The Authors:**

- Is there potential for research on pragmatic quantifiers to be used in quantifier-related inference in 1st-order logic formalism parsed from natural language? I wonder if it is possible to formalize arguments using existential and universal quantification from the range of quantifiers.

**Reasons To Accept:**

- The motivation and problem statement of the paper is clear, and the theoretical background and experimental design to support the authors’ claims are clearly explained, making the text itself highly complete.
- The attempt to limit the range of Generalized Quantifiers using the perspective of pragmatics seems novel, and the QuRe dataset also has a versatility that can be useful in the NLI field where quantifier-related reasoning is required in the future.

**Reasons To Reject:**

- I have no reason to reject.

**Reproducibility:**

4: Could mostly reproduce the results, but there may be some variation because of sample variance or minor variations in their interpretation of the protocol or method.

**Reviewer Confidence:**

4: Quite sure. I tried to check the important points carefully. It's unlikely, though conceivable, that I missed something that should affect my ratings.

---

> ### Author Rebuttal · Authors · 2023-08-29
>
> We thank the reviewer for recognizing and suggesting the impacts of our work. Applying the pragmatic-inference framework of PRESQUE to other inference and reasoning tasks that involve quantifiers (as part of the input) would be an interesting direction. As the reviewer pointed out, some adaptations and attention to the designs are needed, like formalizing the arguments with existential and universal quantification, which are supported by the current 1st-order logic framework.

---

### Official Review · Reviewer_VNWH · 2023-08-07

**Soundness:** 4

**Excitement:**

4: Strong: This paper deepens the understanding of some phenomenon or lowers the barriers to an existing research direction.

**Missing References:**

Bowman at al. 2015a,b, refer to the same publication.

**Paper Topic And Main Contributions:**

The paper introduces an algorithm for interpreting generalized quantifiers (GQs) semantically in terms of percentage ranges (PRESQUE). The interpretation algorithm is theoretically grounded in Gricean pragmatics of modelling a listener who takes into account assumptions about the speaker’s intentions. This is operationalized in a Bayesian approach by using NLI relations between sentences with quantifier and sentences with percentages as well as relative quantifier frequencies.

Second, the paper introduces a new test suite (QuRe) for probing the ability of LLMs to interpret English GQs in terms of percentage ranges. In particular, it consists of 746 sentences that contain one of the following 15 “quantifier words”: {all, generally, most, usually, some, likely, few, little, occasionally, none, seldom, tiny, small, moderate,  large}. The sentences are based on Wikipedia sentences that include percentage information, which were manually rewritten as semantically equivalent sentences with a quantifier expression.

Third, the interpretations of five “classic” GQs (all, most, some, few, no) are crowdsourced in terms of percentage ranges.

The performance of LLMs (and in particular RoBERTa-large) fined-tuned for NLI and of the PRESQUE algorithm are evaluated using (1) the independently existing Herbelot-Vecchio dataset (featuring the five classic GQs), and (2) the newly introduced QuRe dataset (featuring the 15 "quantifier words" above). The results show that the LLMs grasp general differences between GQs scopes but have limited predictive power for percentage ranges. The authors conclude that the pragmatic PRESQUE algorithm produces better percentage predictions than a "literal" NLI based predictor.

**Questions For The Authors:**

Question A: Fleiss kappa for quantifier choice was only 0.37 Fleiss' kappa for 3 annotators.  How often did the annotators choose 3 different quantifiers? Will the release of QuRe include all three annotations for each instance (as in the Herbelot-Vecchio dataset)?

Question B: Will the crowd-sourced percentage interpretations for the 5 quantifiers in the Herbelot-Vecchio by 25 judges be published as a dataset?

Question C:  I don’t understand the relevance of Pandia et al. (2021)’s finding for distance-based metrics (line 514). Is this the correct reference?

**Reasons To Accept:**

The paper makes a contribution to the NLP interpretation of generalized quantifiers (GQs), which are frequent in natural language communication but have been mostly neglected in NLP. GQs are notoriously vague in their meaning but for some applications  (e.g. in human-robot interaction) it is necessary to interpret the quantification in percentages. The paper offers an interesting pragmatically inspired algorithm to determine percentage interpretation on the basis of existing LLMs that are fine-tuned for NLI.

It also introduces the new dataset QuRe of sentences that feature GQs. The QuRe sentences are automatically (GPT-3.5-turbo) annotated with percentage range (derived from the original sentences that included percentage information plus potential modifications such as "about", "at least")  and 3 topic labels. They are also manually annotated with a 3-level “specificity” score which indicates whether the sentence without the quantifier gives partial or full information about the percentages or whether it is indeterminable from the context.

For the 5 quantifiers in the Herbelot-Vecchio dataset percentage interpretations are crowd-sourced from 25 human judges and visually compared to (RoBERTa-large NLI-based ) PRESQUE results.

A random sample of the results on the Herbelot-Vecchio dataset is evaluated by human judges.

The results on the QuRe dataset are evaluated by employing five evaluations measure  that emphasize different properties (HIT@1, MRR, Cross-Entropy, F1@n, and consecutiveness}. Although not good, the results are better than (random seeds) baselines and the generall quantifier strengths correspond to previous hierarchies based on human judgements.

**Reasons To Reject:**

Origin of QuRe und its distribution of GQs: The QuRe dataset is based on Wikipedia sentences that include the target percentages which means that the LLMs have most likely seen the target interpretation in their pre-training data. That decreases the usefulness of the data set.

In addition, the distribution of the quantifier expressions in QuRe is very skewed (26,3 % "some" aka about 196 instances but less than 1% aka 7 or less instances of "likely", "seldom", "occasionally", "none" - the authors only provide relative frequencies). The distribution of percentage scopes is not reported, but doesn't seem to be balanced either.

Linguistic soundness: The set of "quantifier words" is very heterogenous, some are not normally seen as quantifiers: The reference to Srivastava et al. (2018) accounts for the inclusion of frequency adverbs such as "usually", "rarely", but the inclusion of bare adjectives such as "tiny"or "large" without a noun such as "part" or "amount" requires further explanation (I don't think that it is a problem that they are listed in the crowd-sourcing task without explanation, though).

Thoroughness/Informativeness: The quantitative evaluations on QuRe report only averaged values over all quantifiers.

Soundness: The human evaluation on the performance of L0 vs. L1 on the 50 test sentences that were randomly chosen from the Herbelot-Vecchio dataset (10 sentences per each of the 5 quantifiers) evaluated by 40 annotators is not tested for statistical significant differences.  [In favour of the authors' interpretation: If I interpret the numbers in figure 7 correctly the results for "some", "all" and the overall values are statistically significant (based on a chi-square test) and do indeed prefer the output of the PRESQUE L1].

Clarity: The paper works with two datasets with two different sets of "quantifiers". The paper should always indicate which dataset is referred to,  e.g. in all table/figure captions (as it is the case in figure 1 and table 3).

**Reproducibility:**

3: Could reproduce the results with some difficulty. The settings of parameters are underspecified or subjectively determined; the training/evaluation data are not widely available.

**Reviewer Confidence:**

2: Willing to defend my evaluation, but it is fairly likely that I missed some details, didn't understand some central points, or can't be sure about the novelty of the work.

**Typos Grammar Style And Presentation Improvements:**

* 140 “HVD” => the acronym for the Herbelot-Vecchi dataset has not been explicitly introduced in the main text, only in the abstract.
* Section 3: Why are the HVD triples converted into sentences for collecting the wikipedia sentences? Why not just use the concept nouns? (lines 232 ff.) It is confusing that this information is given in section 3 (on QuRe), “Stage 1: Wikipedia sentences collection”.
* Section 3, Statistics: Table 3 should include absolute frequencies of the quantifiers.
* The reason for using HVD concepts as seeds for collecting wikipedia sentences is not explained.
* 223 Clarification needed: “some is used in 25% of sentences” (line 223) vs. figure 1:  proportion of “some” is 26,3% of all quantifiers?
* 288 “Therefore” => maybe rather: This step ensures ….??
* 323: "We have collected 746 ˜ Sq sentences, of which 74% and 13% contain one and two percentage mention(s), respectively." =>  What about the other 13 %?
* Section 5.1. Figure 3:  the number of human interpreters (n=25) should be provided in the caption also the number of instances per quantifier analyzed by PRESQUE.
* Section 5.2 Table 5 => the number of human annotators (n=40) should be provided in the caption (or the actual number for each sentence, if it differed per sentence).
* 512 Which distance-based metric?
* 514 I don’t understand the relevance of Pandia et al. (2021)’s finding for distance-based metrics. Is this the correct reference?
* 521 Why is “only a few school children were recognized..."” fully determinable?
* Table 7: Table 7 should be explicitly anchored in Appendix A (cf. caption of table 1). The header QuRe “sentence” is not clear, because some examples in Table 7 include more than one syntactic sentence.

Typos:

* 321 “and slightly extended” => and was is/slightly extended
* 336 “over 30% of cases” => over 30% of the cases
* 337 “most is selected” => … is selected in
* 339 “18% of sentences” => … of the sentences
* 341 “and 48% indeterminable” => … are indeterminable
* 346 “and shows interesting hierarchies:” => incomplete clause
* 380 “compute following metrics” => compute the following metrics
* 460 “We further calculate L0 result through Eq. 1.” => … the L0 result using Eq. 1”
* check: fine-tuned, cross-entropy for consistent spelling

Bibliography:

* Partly incomplete entries

---

> ### Author Rebuttal · Authors · 2023-08-29
>
> We sincerely appreciate  the reviewer’s detailed feedback and insightful comments. We address the concerns below:
>
> Data leakage of Wikipedia: We use Wikipedia since it is the best source, to the best of our knowledge, with rich percentage mentions that are relatively objective compared to other choices. While we acknowledge the concern about generalization and potential impacts from pretraining exposure, we would like to emphasize that the mere exposure to these sentences doesn't necessarily equate to the direct learning of quantifier semantics (since we recruit annotators to provide the quantified version of those sentences which may not appear in Wikipedia). We hope our work could serve as an exploratory step to collect better benchmarks in the future.
>
> Distribution of generalized quantifiers: We found the imbalance in the use of generalized quantifiers to be related to the natural preference of annotators (e.g. people might be hesitant to provide absolute quantification like ‘none’ or ‘all’), and we would lose a large proportion of the annotations if we plan to balance the quantifiers. We don't find a succinct approach to represent the scope distribution in a figure as the percentage scopes are in diverse values and formats, which is why we apply percentage scope generation based on a few hyperparameters (Section 4 and Appendix E).
>
> Regarding the linguistic soundness, we extended the quantifier set with adjectives based on the limitation of only using adverbs we observed in preliminary experiments and extended the coverage. However, we are aware that the entire set of generalized quantifiers can be much larger and requires future efforts to incorporate. We also decided to leave the choice of nouns that attach to those adjectives to the annotators as it will not significantly change the strength of quantifiers.
>
> Thoroughness/Informativeness: We will attach a more fine-grained discussion in our final version.
>
> Soundness:  The p-values of chi-square test for No, Few, Some, Most, All are 0.258, 0.44, < 1e-6, 0.258 and 0.02. Therefore, as the reviewer correctly guessed, differences in results for Some and All are significant.
>
> Improving clarity: We thank the suggestion to improve clarity and pointing out redundant references. We will make appropriate changes in the final version.
>
> Addressing questions posed by the reviewer:
>
> Question A: In 19% of the cases, all the annotators choose different quantifiers. We will provide all three annotations when releasing the dataset.
>
> Question B: Yes, we will release those annotations.
>
> Question C: Thanks for pointing this out. The correct reference would be Carcassi and Szymanik, 2021. [1]
>
> [1] Fausto Carcassi, Jakub Szymanik, ‘Most’ vs ‘More Than Half’: An Alternatives Explanation, SCiL 2021
>
>
>
> We thank the reviewer for providing the detailed text and reference correction, as well as writing suggestions, we would incorporate them in the final version.
>
>
> We address other comments posed by the reviewer below:
>
> - Section 3: Why are the HVD triples converted into sentences for collecting the wikipedia sentences? Why not just use the concept nouns? (lines 232 ff.) It is confusing that this information is given in section 3 (on QuRe), “Stage 1: Wikipedia sentences collection”.
>
> There are no limitations for using sentences or concepts for the sentence collection. In practice, we found that using the sentences returned more diverse wikipedia entities since one concept might be associated with more than one sentence.
> Section 3, Statistics: Table 3 should include absolute frequencies of the quantifiers.
> The absolute strengths (if that is what the question means) of quantifiers cannot be determined without hyperparameters like granularity and window size since the percentage mentions usually refer to scopes (Table 2 includes some examples).
> The reason for using HVD concepts as seeds for collecting wikipedia sentences is not explained.
> Since we use an off-the-shelf entity linking library, we need a seed corpora to start the searching, and therefore we choose the HVD concepts as the start. However, the procedure can be easily substituted with other resources.
>
> - 223 Clarification needed: “some is used in 25% of sentences” (L223) vs. figure 1: proportion of “some” is 26,3% of all quantifiers?
>
> We missed the word ‘over’, thanks for pointing it out.
>
> - 323: "We have collected 746 ˜ Sq sentences, of which 74% and 13% contain one and two percentage mention(s), respectively." => What about the other 13 %?
>
> The other sentences include more than two percentage mentions.
>
> - 512 Which distance-based metric?
>
> The distance-based metric we use is minimal scope distance (MSD) which we elaborate in Appendix G. We would rephrase it in a more explicit way in L512 in the final version.
>
> - 514 I don’t understand the relevance of Pandia et al. (2021)’s finding for distance-based metrics. Is this the correct reference?
> Thanks for pointing it out. We have included the correct reference in the answer to Question C521 Why is “only a few school children were recognized..."” fully determinable?
>
> Thanks for pointing out our mistake. The original sentence was “In the 2017 Dutch study only 2 out of the total 27 school children (7%) recognized that the website was a hoax.” (which also shows up in Table 8) and we mistakenly placed the corresponding sentence with the quantifier in the text. We would correct this in the final version.

---

### Official Review · Reviewer_dNQH · 2023-08-11

**Soundness:** 4

**Excitement:**

4: Strong: This paper deepens the understanding of some phenomenon or lowers the barriers to an existing research direction.

**Paper Topic And Main Contributions:**

This paper introduced a new corpus QuRe for quantifier semantics inferences. Given quantifiers in each sentence, it is annotated with a percentage scope. A framework PRESQUE combing natural language inference with Rational Speech Acts framework is proposed to predict the percentage scopes for quantifiers. Detailed annotation procedures and statistical analysis are presented. Preliminary experimental results show that PRESQUE can perform better than a literal listener on various evaluation

**Questions For The Authors:**

1. Annotation clarification:

    a. There are certain operators in Table 2 without a percentage upper or lower bound, such as “over”, and “not exceeding”. For these operators, how the percentage scopes are determined?

    b. Could you elaborate more on the design of the window size? And how does the window size influence the operators and the percentage scope during annotation?

2. Evenly spaced model output:

    a. For different quantifiers, the corresponding ranges can vary a lot as shown in Table 3. However, the percentage spectrum is divided into evenly spaced intervals. The model’s categorical output may influence the model’s inference on certain quantifiers that can cover a large range.

    b. Can the framework be adapted to a value regression setting, which may better fit the way that humans interpreted it?

3. Framework clarification: the RSA framework is good at solving tasks that need to speak or comprehend sentences heavily dependent on the current dialogue context. Could you further clarify the following in your task:

    a. How the dialogue context influences the quantifier semantics influences in this task?

    b. And, how this influences in (a) reflected in terms of equations (1-4)? From my understanding, the pragmatic listener in equation (2) only takes additional information about the word frequency compared with the literal listener but not grounds in the current sentence context.

    c. How the entailment score is computed?

**Reasons To Accept:**

1. The paper smartly designed a new corpus with quantifiers mapped to a percentage scope which be easily evaluated for quantifier semantics inference.
2. The authors presented the annotation and data statistics in detail. A suite of comprehensive evaluations is conducted including the comparison of the distributions between human and model interpretation, automatic metrics, consecutiveness, etc.

**Reasons To Reject:**

Overall this paper presented a thorough introduction to data collection, analysis, and experimental results. I have several clarification questions regarding the framework design and annotation procedure. Please check below.

**Reproducibility:**

3: Could reproduce the results with some difficulty. The settings of parameters are underspecified or subjectively determined; the training/evaluation data are not widely available.

**Reviewer Confidence:**

3: Pretty sure, but there's a chance I missed something. Although I have a good feel for this area in general, I did not carefully check the paper's details, e.g., the math, experimental design, or novelty.

---

> ### Author Rebuttal · Authors · 2023-08-29
>
> We thank the reviewer for the thorough feedback. We address the questions raised by the reviewer below:
>
> Question 1.a In general, all the percentage scopes are bounded between 0 and 100%. Moreover, we apply several rules to generate the percentage scope using few hyperparameters (the detailed production rules are listed in Table 10 of Appendix E).
> Question 1.b The window size is separate from the annotation, the annotators are only required to provide a quantified sentence and the specificity. The window size is introduced by us to help frame the original ambiguity in the percentage expressions (e.g. around 10%) in a way a NLI model like RoBERTa is able to evaluate.
>
> Since the original sentences (with percentage mentions) in QuRe include different percentage expressions (and some of them like ‘around 10%' do not refer to exact percentage values), they need to be formatted in a way the model with unified granularity is able to handle. Therefore we introduce the window size to convert the percentage expressions into percentage scope with more explicit boundaries to meet the setup.
>
> Question 2.a The final percentage scope from the model is aggregated from the top categorical outputs (L461-463). Therefore, with the granularity and the number of top outputs as hyperparameters, the model is able to output different percentage scopes without training instances.
>
> Question 2.b Our framework can also be extended to accommodate the regression setting, where we train models to output percentage values corresponding to a quantifier. However, since the same quantifier may not correspond to a single percentage value in different contexts, we may have to develop other approaches to make model results interpretable to humans (for example, learning a distribution instead of single values).
> In this regard, we would like to emphasize the broader impact of the data/annotations collected by us, as it can be adapted into various formats for different techniques of modeling quantifier semantics. We encourage the community to explore such avenues in the future to further expand the capabilities of PRESQUE.
>
> Question 3.a Specific topic that the context involves can skew the quantifier choices, for example, the threshold of ‘large’ in mortality or the salinity of water bodies can be lower than other topics (we list several annotated sentences with their original counterpart in wikipedia):
>
> “The NCDUs suffered 31 killed and 60 wounded, a large casualty rate.”
> “The NCDUs suffered 31 killed and 60 wounded, a casualty rate of 52%.”
>
> “The Dead Sea, known for its large salinity levels, is really a salt lake.”
> “The Dead Sea, known for its salinity levels between 30 and 40%, is really a salt lake.”
>
> In comparison, sentences from other topics may look like,
> “The bitumen content is large (soluble in carbon disulphide), with a penetration value near to zero and a softening point (ring and ball) around 120 °C.”
> “The bitumen content varies from 83% to 92% (soluble in carbon disulphide), with a penetration value near to zero and a softening point (ring and ball) around 120 °C.”
>
> Question 3.b $S_{0}(q|p)$ includes the contextual understanding of the quantifier from the NLI framework. And the computation of $S_{0}(q|p)$ (Eq. 3) is different from $L_{0}(p|q)$ (Eq. 1) by taking a different order of inputs. We explain this choice in the fourth paragraph of the introduction (L70) addressing the capability of current NLI frameworks in handling ambiguity and elaborate it in Appendix C. Also, we can notice that the improvement of PRESQUE over the L_0 is different among different specificities, which is affected by the comprehension of context.
>
> Question 3.c The entailment score is computed by setting the premise and hypothesis as $\tilde{S_{q}}$ and $\tilde{S}_{p}$ respectively (or the other way around) in L168-188, the output scores are the scores corresponds to entailment/netural/contradiction respectively that sum up to be 1.

---

### Meta-Review · Area_Chair_w4ve · 2023-09-18

**Recommendation:** 4

**Metareview:**

This paper conceptualizes generalized quantifier semantics as percentage ranges, and introduces 1) an annotated dataset of generalized quantifiers in Wikipedia sentences, and 2) a model framework combining NLI and RSA to predict percentage ranges. The paper shows that the proposed pragmatic model outperforms a “literal” RoBERTa-large model fine-tuned on NLI.

The reviewers express that the paper is clear and well-motivated, addressing a worthwhile and understudied topic, and contributing a valuable corpus and an interesting pragmatically-driven model. Some concerns are raised about clarity and soundness, but these have largely been addressed in the discussion period. Overall, this paper will likely be a solid, interesting, and novel contribution to the conference.

---

### Decision · Program_Chairs · 2023-10-07

**Decision:**

Accept-Main

**Comment:**

This paper conceptualizes generalized quantifier semantics as percentage ranges, and introduces 1) an annotated dataset of generalized quantifiers in Wikipedia sentences, and 2) a model framework combining NLI and RSA to predict percentage ranges. The paper shows that the proposed pragmatic model outperforms a “literal” RoBERTa-large model fine-tuned on NLI.

The reviewers express that the paper is clear and well-motivated, addressing a worthwhile and understudied topic, and contributing a valuable corpus and an interesting pragmatically-driven model. Some concerns are raised about clarity and soundness, but these have largely been addressed in the discussion period. Overall, this paper will likely be a solid, interesting, and novel contribution to the conference.